

# Impact of improved representation of VOC emissions and production of NOₓ reservoirs on modeled urban ozone production

Katherine R. Travis[1], Benjamin A. Nault[2,3], James H. Crawford[1], Kelvin H. Bates[4], Donald R. Blake[5], Ronald C. Cohen[6,7], Alan Fried[8], Samuel R. Hall[9], L. Gregory Huey[10], Young Ro Lee[11], Simone Meinardi[5], Kyung-Eun Min[12], Isobel J. Simpson[5], Kirk Ullman[9]

[1]NASA Langley Research Center, Hampton, VA, USA
[2]CACC, Aerodyne Research, Inc., Billerica, MA, USA
[3]Department of Environmental Health and Engineering, Johns Hopkins University, Baltimore, MD, USA
[4]NOAA Chemical Sciences Laboratory, Earth System Research Laboratories, and Cooperative Institute for Research in Environmental Sciences, University of Colorado, Boulder, CO 80305, USA
[5]Department of Chemistry, University of California, Irvine, Irvine CA USA 92697
[6]Department of Chemistry, University of California, Berkeley, CA, USA
[7]Department of Earth and Planetary Science, University of California, Berkeley, CA, USA
[8]INSTAAR, University of Colorado, Boulder, CO, USA
[9]Atmospheric Chemistry Observations & Modeling Laboratory, NCAR, Boulder, CO, USA
[10]School of Earth and Atmospheric Sciences, Georgia Institute of Technology, Atlanta, GA, USA
[11]Division of Geological and Planetary Sciences, California Institute of Technology, Pasadena, CA, USA
[12]School of Environmental Sciences and Environmental Engineering, Gwangju Institute of Science and Technology, Gwangju, South Korea

*Correspondence to*: Katherine R. Travis (katherine.travis@nasa.gov)

**Abstract.**

The fraction of urban volatile organic compounds (VOC) emissions attributable to fossil fuel combustion has been declining in many parts of the world, resulting in a need to better constrain other anthropogenic sources of these emissions. During the National Institute of Environmental Research (NIER) and National Aeronautics and Space Administration (NASA) Korea-United States Air Quality (KORUS-AQ) field study in Seoul, South Korea during May-June 2016, air quality models underestimated ozone, formaldehyde, and peroxyacetyl nitrate (PAN) indicating an underestimate of VOCs in the emissions inventory. Here, we use aircraft observations interpreted with the GEOS-Chem chemical transport model to assess the need for increases in VOC emissions. We find that the largest increases are attributable to compounds associated with volatile chemical products, liquefied petroleum gas (LPG) and natural gas emissions, and long-range transport. Revising model chemistry to better match observed VOC speciation together with increasing model emissions of underestimated VOC species increased calculated OH reactivity by +2 s⁻¹ and ozone production by 2 ppb hr⁻¹. Ozone increased by 6 ppb below 2 km and 9 ppb at the surface, and formaldehyde and acetaldehyde increased by 30% and 120% aloft, respectively, all in better agreement with observations. The larger increase in acetaldehyde was attributed to ethanol emissions which we found to be as important for ozone production as isoprene or alkenes. The increased acetaldehyde largely resolved the model PAN bias. The need for additional unmeasured VOCs however was indicated by a remaining model bias of -1 ppb in formaldehyde and 57% and 52%




underestimate in higher peroxynitrates (PNs) and alkyl nitrates (ANs), respectively. We added additional chemistry to the model to represent an additional six PNs from observed VOCs but were unable to account for the majority of missing PNs.

However, four of these PNs were modeled at concentrations similar to other commonly measured PNs (>2% of PAN) indicating that these should be measured in future campaigns. We hypothesize that emissions of oxygenated VOCs (OVOCs) such as $\geq$C5 aldehydes from cooking and/or alkenes associated with volatile chemical products could produce both PNs and ANs and improve remaining model biases. Emerging research on the emissions and chemistry of these species will soon allow for modeling of their impact on local and regional photochemistry.

## 1 Introduction

Ozone pollution in urban areas may be limited by availability of nitrogen oxides ($NO_x$) or volatile organic compounds (VOCs). Emissions inventories of VOCs are more difficult to estimate than for $NO_x$ due to the large number of compounds that must be included and the lack of measurements of many of these species. In general, VOC emission inventories have been shown to perform poorly around the globe against observations (von Schneidemesser et al., 2023; Rowlinson et al., 2023). Non-

combustion sources such as volatile chemical product (VCP) emissions are becoming a larger fraction of urban VOC emissions in many cities and have only recently become a focus of emissions inventory development (McDonald et al., 2018; Coggon et al., 2021). In some cities, non-combustion anthropogenic emissions of VOCs from diverse products and processes may be equivalent to or greater than motor vehicle emissions (Khare and Gentner, 2018; McDonald et al., 2018; Simpson et al., 2020). This has implications for simulating ozone production in cities where ozone production is VOC-limited including several U.S.

cities (Koplitz et al., 2021) and across much of East Asia (Lee et al., 2021).

The joint National Institute of Environmental Research (NIER) and National Aeronautics and Space Administration (NASA) Korea-United States Air Quality (KORUS-AQ) field study in May-June 2016 (Crawford et al., 2021) presented an opportunity to better constrain VOC emissions in East Asia, with a focus on South Korea and eastern China. Observations included airborne

measurements using the NASA DC-8 aircraft and ground-based measurements at the Olympic Park supersite in Seoul. The suite of models run for this campaign generally underestimated ozone and formaldehyde, a common oxidation product of VOCs, suggesting underestimation of VOCs in the emissions inventory (Park et al., 2021). Models also underestimated peroxyacetyl nitrate (PAN), a product of VOC oxidation in the presence of $NO_x$ and a reservoir for ozone precursors that can be transported long distances from source regions (Wolfe et al., 2007; Bertram et al., 2013). This has implications for the

ability of models to attribute the relative impact of upwind versus local emissions on downwind pollution.

Several studies have discussed model biases in ozone and formaldehyde concentrations in East Asia. Kim et al. (2022) showed that modeled VOCs significantly underestimated the overall OH reactivity during KORUS-AQ indicating that modeled ozone production was underestimated. Gaubert et al. (2020) found that persistent underestimates in modeled carbon monoxide (CO)





in East Asia were partially responsible for the modeled ozone underestimate. Miyazaki et al. (2019) assimilated multiple species observed from satellite observations including CO and PAN into a model which improved performance for ozone but not formaldehyde, indicating that the assimilation was missing a correction for underestimated VOCs. Choi et al. (2022) used satellite formaldehyde observations to improve modeled VOCs and ozone, but biases still remained, likely due to remaining errors in CO emissions and VOC emission speciation. Using airborne remote sensing formaldehyde data, Kwon et al. (2021)

found differences in the KORUS-AQ anthropogenic VOC emissions inventory of up to a factor of 6.9 although they were limited by their sparse observational dataset.

Studies have paid less attention to model underestimates of specific VOCs, errors in model VOC speciation, and to the underestimate of PAN. For example, ethanol, a major PAN precursor (Fischer et al., 2014), was not included in the KORUS-

AQ emissions inventory but has been measured at high concentrations in East Asia (Kim et al., 2016; Wu et al., 2020). Yang et al. (2023) performed modeling of South Korea with an inventory for volatile chemical products and found that this source greatly increased simulated ethanol, methanol, and acetone. However, this study's scope did not include a detailed comparison against observations or their oxidation products such as acetaldehyde or PAN. In addition to missing PAN precursors, models generally simulate few other peroxynitrate (PN) species. Lee et al. (2022) used a box model of observed VOCs during KORUS-

AQ to estimate that PNs other than PAN and peroxypropionyl nitrate (PPN) could be up to 40% of total peroxy acyl nitrates ($\sum PNs$), in contrast to previous findings that this fraction is less than 20% (Wooldridge et al., 2010). In a study of a petrochemical region of Korea, the major PNs identified by Lee et al. (2022) were produced from oxidation of 1,3-butadiene and glycolaldehyde but this chemistry is not included in most atmospheric chemistry models. Alkyl nitrates (ANs) are another reservoir of $NO_x$ produced during VOC oxidation that competes with ozone production in urban regions and can serve as an

ozone source downwind (Perring et al., 2010; Farmer et al., 2011). Total ANs ($\sum ANs$) were underestimated by -50% by models during KORUS-AQ (Park et al., 2021).

Nault et al. (2024) showed that in the Seoul Metropolitan Area (SMA), $\sum PNs$ accounted for 33% of observed $NO_z$ ($\equiv$ $\sum PNs + \sum ANs + HNO_3 + aerosol\ nitrate$), where only 50% of $\sum PNs$ was PAN. $\sum ANs$ were 10% of observed $NO_z$ with

only ~20% of $\sum ANs$ accounted for by speciated observations. This is similar to the finding of Kenagy et al. (2021) that their model could only account for 30% of $\sum ANs$ across all data (not just the SMA) collected during KORUS-AQ. Nault et al. (2024) found that between 20 – 70% of the PAN precursor budget was attributable to ethanol, depending on proximity to the emissions source. Additional higher peroxy acyl nitrates (PNs) were shown to be produced by observed VOCs including glycolaldehyde, aromatics, monoterpenes, 1,3-butadiene, and methyl ethyl ketone (MEK). Here, we use the KORUS-AQ

aircraft observations of ozone-$NO_x$-VOC chemistry interpreted with the GEOS-Chem chemical transport model to assess underestimated modeled VOCs on a species-by-species basis, and determine the impact on model biases in ozone, formaldehyde, $\sum PNs$, and $\sum ANs$. We build on the findings of Nault et al. (2024) to add additional chemistry to the model to



form higher PNs from observed VOCs and assess indicators of additional missing sources of VOCs to close remaining biases in ozone, formaldehyde, $\sum PNs$, and $\sum ANs$.


## 2 Observations during KORUS-AQ

The KORUS-AQ campaign took place from May 1 to June 10, 2016, in Seoul, South Korea (Crawford et al., 2021). KORUS-AQ was a joint field campaign organized by South Korea's National Institute of Environmental Research (NIER) and the United States National Aeronautics and Space Administration (NASA). The campaign included 20 flights using the NASA

DC-8 aircraft which performed 55 missed approaches at multiple times per day over the heavily instrumented Olympic Park supersite in Seoul. Ground-based ozone and $NO_2$ observations were available from the NIER's AirKorea monitoring network including locations near Olympic Park. Crawford et al. (2021) provide a full listing of all observations made during KORUS-AQ. Table 1 describes the aircraft and ground observations used in this work.

KORUS-AQ did not measure ethanol concentrations either from aircraft or ground-based instruments. During the MAPS-Seoul campaign in May-June 2015, Kim et al. (2016) measured concentrations at Olympic Park of methanol and ethanol of 11.1 ppb and 3.9 ppb, respectively. Wu et al. (2020) measured methanol at 11.4 ppb and ethanol at 5.6 ppb at Guangzhou in China in September-November 2018. We used these observations to estimate that ethanol was equivalent to methanol/2.5 as was done in Schroeder et al. (2020). In the U.S., similar ratios were observed in the Northeast (Sommariva et al., 2011), and

even higher levels of ethanol than methanol have been observed in California (de Gouw et al., 2018).

## 3 Modeling Setup and Improvements

We used the GEOS-Chem chemical transport model version 13.4.0 (10.5281/zenodo.6511970) described in (Travis et al., 2022) with modifications described below to VOC speciation and chemistry. Kim et al. (2022) showed that the model had errors in OH reactivity during KORUS-AQ of ~34% for $\geq$C4 alkanes (ALK4) due to lumping. Alkanes with larger carbon

numbers (e.g., n-hexane) are more reactive than ALK4 which is parameterized in the model as a butane/pentane mixture (Lurmann et al., 1986). We added a new lumped species for $\geq$C6 alkanes, ALK6, with associated chemistry including the formation of a lumped alkyl nitrate species according to (Lurmann et al., 1986).

GEOS-Chem includes chemistry for the aromatic species benzene, toluene, and xylenes. Significant effort was made during

KORUS-AQ to improve emissions estimates of these species, particularly toluene, which was determined to be a major contributor to chemistry in the SMA (Schroeder et al., 2020; Simpson et al., 2020). However, emissions improvements did not take into consideration the fraction of ethylbenzene (EBZ) or trimethylbenzene (TMB) in modeled aromatic emissions, both of which are more reactive than benzene or toluene. Similarly, the styrene (STYR) fraction of emitted olefines (alkenes) was not considered. KORUS-AQ measurements included observations of EBZ, TMB, and STYR (Simpson et al., 2020). Emitted



olefins also include a fraction of 1,3-butadiene (C4H6), especially from petrochemical facilities in western Korea, which were identified by (Lee et al., 2022) as a source of PNs through production of peroxyacrylic nitric anhydride (APAN). 1,3-Butadiene mixing ratios were over 3 ppb near the petrochemical facilities although the maximum levels in Seoul were much lower at <200 pptv (Simpson et al., 2020). We added chemistry for STYR, EBZ, and TMB from Bates et al. (2021) and for C4H6 from the MCMv3.3.1 (Jenkin et al., 1997; Saunders et al., 2003). Finally, we include updated monoterpene chemistry that is a

condensation of the mechanism from the MCMv3.3.1 (Saunders et al., 2003) that includes production of aldehydes that could form PNs such as pinonaldehyde. Tables S1 and S2 provide the new model chemistry implementation.

Table 2 shows the VOC emission species in the KORUSv5 inventory, which was developed by Konkuk University for the campaign, speciated according to the SAPRC99 mechanism. We translated this mechanism to the GEOS-Chem model for the

base chemistry, and updated chemistry, according to (Carter, 1999) with the exception of ARO1, which was speciated based on observations as discussed in Travis et al. (2022). We specifically re-speciated base model emissions for ALK4 into ALK4 and ALK6, emissions for PRPE into PRPE, C4H6, and STYR, emissions for benzene and toluene into benzene, toluene, and EBZ, and emissions of xylenes into xylenes and TMB. The KORUSv5 inventory does not include ethanol emissions, which we took from the Community Emissions Data System (CEDS) inventory described in (McDuffie et al., 2020).


Figure S1 shows model and observed mean vertical profiles in the SMA for the key species identified by Schroeder et al. (2020) for ozone production (C7+ aromatics, isoprene, alkenes, methanol), additional species identified by Nault et al. (2024) for PAN and PN production (ethanol, monoterpenes, methyl ethyl ketone (MEK)), and CO, identified as an additional source of model errors in ozone chemistry during the campaign (Gaubert et al., 2020). Fried et al. (2020) found that emissions of the

top producers of formaldehyde, particularly propene and ethene were underestimated by the KORUSv5 emissions inventory over the industrial area to the southwest of Seoul. We did not find these species to be underestimated in the SMA. Figure S1 shows that model underestimates in CO and VOCs range from more than -70% (methanol, ethanol, propane, MEK, monoterpenes) to -30 to -50% (acetone, ALK4, benzene, CO, acetylene, ethylbenzene, toluene, xylenes). We applied scaling factors to the individual KORUSv5 VOC emissions over South Korea until modeled concentrations matched observations.

Ethanol was scaled from the emissions in the CEDS inventory. For species with lifetimes long enough to be transported from upwind (e.g., acetone, CO) some of the model bias may be due to underestimated emissions from other countries. Here, we only scaled South Korean emissions given the lack of constraints on emissions upwind. Table 3 provides the applied scale factor for each species. Figure S1 shows the model with applied scaling factors for each scaled species which shows significantly improved comparison with observations. Given the difficulty of achieving perfect scaling factors to achieve model

agreement across this large suite of VOCs, and the likelihood that some scale factors are needed for upwind emissions, we present these scale factors not as exact values that need to be implemented in emissions inventories but rather strong indicators of missing sources that need further study.



The largest scaling factor (650x) was required for methanol (Table 3), which averaged 20 ppb in the observations but only 2
ppb in the base model. Given the relatively long lifetime of methanol (~5 days), we expect that this very large anthropogenic
scaling factor is needed to account for underestimated emissions both upwind in China and in South Korea and for the
contribution of biogenic methanol to model concentrations. According to Simpson et al. (2020), methanol in the SMA
correlated well with ethylbenzene suggesting that the missing anthropogenic source is from solvent emissions. Underestimated
solvent emissions may also be the reason for underestimated model xylenes and ethylbenzene (Simpson et al., 2020). The
second largest scaling factor was for monoterpenes (450x) which averaged 50 ppt in the observations but 15 ppt in the base
model. This large scaling factor is needed to account for the minimal anthropogenic emissions in the model as the base model
concentration is driven almost entirely by biogenic emissions. Monoterpenes, particularly limonene, have been shown to have
a large source from fragranced VCPs in some urban areas (Coggon et al., 2021; Peng et al., 2022; Wernis et al., 2022). Large
scaling factors were also required for acetone (85x), MEK (70x), and ethanol (40x). Acetone is a common ingredient in paint
thinners. Ethanol is an ingredient in many VCPs (pesticides, personal care products, cleaning, coatings, adhesives, inks
(Gkatzelis et al., 2021; McDonald et al., 2018)) and cooking (Arata et al., 2021). Ethanol and monoterpenes were also large
sources of missing model VOC reactivity and ozone production in a study in Los Angeles and Las Vegas (Zhu et al., 2023)
suggesting that inventories generally have difficulty capturing VCP emission levels. MEK is also a common VCP marker
(McDonald et al., 2018). Underestimated model propane and C3/C4 alkanes (ALK4) may be attributable to underestimated
liquified petroleum gas (LPG) or natural gas emissions, which is used for residential heating and cooking and some vehicles
in South Korea (Simpson et al., 2020). The modeled underestimate of the long-lived combustion tracers carbon monoxide
(CO), ethyne ($C_2H_2$), and benzene (Simpson et al., 2020) is expected given the general underestimate in CO identified in East
Asia by Gaubert et al. (2020) and the expectation that underestimated emissions in South Korea likely reflect underestimated
emissions across East Asia. Overall, we find that the KORUSv5 emissions inventory appears to underestimate VCP and LPG
emissions likely both in South Korea and upwind in East Asia.

## 4 Impact of improved representation of VOCs on model photochemistry

Increasing model VOCs to better match observations resulted in improved representation of calculated OH reactivity (cOHR).
We determined cOHR for the suite of observed VOCs, $CH_4$, and CO (Table 1), and included the non-measured VOC oxidation
products calculated from the F0AM box-modeling results of the SMA from Nault et al. (2024). Figure 1a compares this cOHR
(observed CO+$CH_4$+VOCs+F0AM oxidation products) to modeled cOHR as a function of altitude below 2 km for the SMA.
Figure 1 includes a calculation of the estimated missing reactivity (grey dashed line) which is discussed further in Section 6.
Data is restricted to after 11 am local time according to Nault et al. (2024) to ensure that the aircraft data is minimally affected
by rapid changes in boundary layer growth, the nighttime residual layer, and titration of $O_3$ by NO. The base model cOHR was
only 4.9 s$^{-1}$ in the lowest altitude bin (~0.2km) compared to 9.2 s$^{-1}$ in the observations. Unmeasured VOC oxidation products
(calculated by F0AM) made up 19% of the cOHR, while in GEOS-Chem this was only 11%. Increasing modeled emissions





of CO and VOCs (called "scaled VOCs hereafter) in South Korea as described in Section 2 increased modeled cOHR by 2.4 $s^{-1}$ to 7.3 $s^{-1}$ and significantly reduced this bias.

The increase of model cOHR had a significant impact on ozone production. On average, modeled OH (Fig. 1b) was largely

unchanged but ozone production increased by 2 ppb $hr^{-1}$ (Fig. 1c). This implies that the increased OH sink from VOCs was balanced by increased OH production from recycling ($HO_2$ + NO) and/or photolysis of VOC oxidation products (e.g., formaldehyde). The average model overestimate of ~30% in OH below 2 km is partially attributed to insufficient model resolution and dilution of $NO_x$ given that increasing resolution to ~7 km largely resolved the model bias in OH in Jo et al. (2023). The modeled average net production of ozone + $NO_2$ ($PO_x$) was overestimated compared to observationally constrained

$PO_x$ (Fig. 1c) which was calculated as described in Nault et al. (2024) using steady-state assumptions and observed VOC concentrations which we also attribute to insufficient model resolution and the modeled OH overestimate.

Figure 2 shows the same observations as in Figure 1 but as a function of $NO_x$ concentration instead of altitude. The modeled OHR is closer to the observations at lower $NO_x$ concentrations in part due to the smaller influence of VOC oxidation products that are underestimated against the F0AM box modeling results. Below ~8 ppb $NO_x$, increasing VOCs reduced OH by up to

1E6 molec $cm^{-3}$ while at higher $NO_x$ OH increased by as much as +0.3E6 molec $cm^{-3}$ (Fig. 2b). The larger absolute reduction in OH at lower $NO_x$ is consistent with the main sinks being $HO_2$ + $RO_2$ to form organic peroxides and $HO_2$ + $HO_2$ to form hydrogen peroxide (Nault et al, 2024). Modeled $PO_x$ increased by up to 2.6 ppb $hr^{-1}$ at higher $NO_x$ under VOC-limited conditions. The model does not fully capture the behavior of $PO_x$ in the observations which clearly show a shift from increasing

to decreasing $PO_x$ with increasing $NO_x$ at approximately 6 ppb $NO_x$. Presenting only average conditions for the SMA in Fig. 1 masks the two differing chemical regimes present in the observations. We hypothesize that model difficulty in capturing these two regimes is also due to insufficient model resolution.

Scaled VOC emissions as described in Section 2 increased VOC-oxidation products such as formaldehyde and acetaldehyde

in addition to ozone production. Figure 3 shows that formaldehyde and acetaldehyde were biased by -47% and -67%, respectively in the base model, and increased by 30% and 120% after increasing the VOC emissions. The larger impact of increased VOC emissions on acetaldehyde is attributable to the significant increase in modeled ethanol (Table 3), a key precursor. Unlike in the F0AM box-modeling work in Nault et al. (2024), we were unable to fully reproduce observed formaldehyde in GEOS-Chem after scaling VOCs and CO. Section S1 describes a sensitivity test in F0AM where we reduced

the VOCs and CO in F0AM by the base model bias (Table S3). This test illustrated tha important impacts of insufficient VOCs and CO in Seoul are 1) underestimated loss of OH by reaction with VOCs and CO, 2) missing production of ozone from VOCs and CO through $HO_2/RO_2$ + NO, 3) underestimated conversion of NO to $NO_2$ by ozone which results in underestimated loss of OH by reaction with $NO_2$. These impacts help explain the consequences of underestimated model VOCs and CO on successfully simulating overall ozone photochemistry. We explain the remaining formaldehyde low bias in GEOS-Chem by





the fact that 1) the model underestimates reactivity of intermediate species which provide additional HCHO production, and conversion of NO to $NO_2$ by $HO_2$ and $RO_2$; 2) we are not able to achieve perfect agreement with VOC observations, which is possible with F0AM; and 3) model ozone remains underestimated leading to and thus insufficient loss of OH and a HCHO lifetime against OH that is too short. This $NO_2$ underestimate is made worse by insufficient model resolution.

Despite the model shortcomings listed above, after scaling VOCs and CO, model ozone increased by +6 ppb, reducing the model underestimate from -21 ppb to -15 ppb (Fig. 3c). Previous work attributed underestimated modeled ozone to underestimated influx of stratospheric ozone (Park et al., 2021) or photolysis of particulate nitrate (Colombi et al., 2023; Yang et al., 2023) although this latter mechanism is uncertain (Shi et al., 2021; Gen et al., 2022; Sommariva et al., 2023). Model resolution is unlikely the primary factor as a similar ozone bias was found in a recent study with a range of resolutions in
simulating KORUS-AQ ozone observations in the SMA (Jo et al., 2023). Here, we found that a significant fraction of this underestimate is due to underestimated VOCs. Additional bias could be attributable to underestimated ozone production upwind, as here we only increased VOC emissions in South Korea. Future work should assess how much underestimated VOCs in the rest of East Asia (e.g., China) could contribute to underestimated free tropospheric and surface ozone in models over South Korea. Finally, as formaldehyde is still underestimated in the model by -1 ppb, additional unmeasured VOCs could
be present, and this possibility is further discussed in Section 5. We do not anticipate that the formaldehyde bias is entirely caused by the modeled OH overestimate as increased resolution and improved OH did not resolve the model bias in Jo et al. (2023) implying that underestimated VOC emissions may be the root cause.

    Schroeder al. (2020) identified C7+ aromatics (toluene, xylenes, ethylbenzene) as being the largest driver of ozone production
(32%) in the SMA followed by isoprene and alkenes (14-15%), but a sensitivity test for the impact of ethanol was not included in their calculations. We performed two simulations, one removing ethanol and the other removing C7+ aromatics, over South Korea. Figure S2b shows that reducing ethanol resulted in a 50% reduction in acetaldehyde while removing C7+ aromatics only reduced acetaldehyde by ~10%. C7+ aromatics had a larger impact than ethanol on formaldehyde (8% vs. 6%, Fig. S2a). Removing C7+ aromatics reduced ozone by -3.4 ppb while removing ethanol reduced ozone by -1.7 ppb (Fig. S2c). This result
implies that ethanol was similar in importance during the campaign to isoprene or alkenes which produced ozone at approximately half the rate as C7+ aromatics in Schroeder et al. (2020). Both ethanol and C7+ aromatics had a similar impact on PAN (15-20%, Fig. S2f). Ethanol produces PAN through production of acetaldehyde. C7+ aromatics produces PAN from methylglyoxal and had a larger impact on the NO to $NO_2$ ratio (Fig. S2d). Removing ethanol had minimal impact on OH as the reduced loss from both ethanol and acetaldehyde appeared to be balanced by decreased recycling from $HO_2$ + NO (not
shown), while reducing C7+ aromatics decreased OH by -10% from reduced recycling.

    Figure 4a shows the diurnal cycle of surface ozone at the Olympic Park supersite where scaled VOCs increased modeled ozone by +9 ppb, largely reducing the midday bias. However, the model overestimated the average ozone in the 15 AirKorea sites





contained within the model grid box (shown in Fig. 1b, Travis et al. (2022)). This is likely attributable to the model's inability to resolve the $NO_2$ levels in the grid-box (Fig. 4b) due to insufficient model resolution (e.g, Jo et al., 2023) which were on average higher than at Olympic Park resulting in suppressed ozone production. There was a large gradient of 16 ppb between the observed ozone from aircraft (Fig. 3c) and the daytime (11-16 LT) average surface ozone from the EPA monitor on flight days (Fig. 4a). This gradient would be even larger comparing to the grid-box average ozone value which is lower as discussed above. The model shows no gradient. Park et al. (2021) attributed the strong ozone gradient observed in the boundary layer to

suppressed $HO_x$ at high $NO_x$ and increased ozone destruction by NO and VOCs. Insufficient model resolution here (~25 km) may be the cause of the lack of model gradient below 1 km as Jo et al. (2023) showed a decreasing gradient in the lowest 1 km at high resolution (< 14 km) compared to an increasing gradient at low resolution (> 56 km).

Figure 5a shows model maps of maximum daily 8-hour average (MDA8) surface ozone for the campaign (May 1st-June 10th,

2016). Suppressed ozone production in VOC-limited conditions is evident in the model in the SMA and to a lesser extent in Busan on the southeastern coast where MDA8 ozone is significantly lower than the surrounding areas. MDA8 ozone over both areas increased by as much as +10 ppb (Fig. 5b) due to scaled VOCs over South Korea.

## 5 Model simulation of peroxy acyl nitrates ($\sum PNs$)

Peroxyacetyl nitrate (PAN) is the simplest and most abundant peroxy acyl nitrate (PN). It is produced in the SMA largely from

ethanol, isoprene, C8 aromatics, toluene, and MEK oxidation (Nault et al., 2024). As discussed in Section 3, all these species except for isoprene were increased to better match observations during KORUS-AQ (Fig. S1, Table 3). Figure 3d shows that this scaling reduced the modeled PAN underestimate from -50% to -23%. PAN formation is sensitive to the ratio of $NO/NO_2$ (Nihill et al., 2021). The modeled $NO/NO_2$ ratio below 0.2 km decreased from 0.27 to 0.25 when the VOC scaling was applied due to increased conversion of NO to $NO_2$ (Figure S3c) but was still overestimated compared to the observed ratio (0.20) likely

contributing to the remaining model bias. This may be due to the model's inability to resolve higher levels of $NO_2$ as well as the need for further radical sources to increase NO to $NO_2$ conversion.

Observed PAN averaged 1.1 ppb below 2 km (Fig. 3d) and made up only approximately 50% of $\sum PNs$ which averaged 2.5 ppb (Fig. 3d+e). Peroxypropionyl nitrate (PPN) averaged 80 ppt, or 7% of PAN. Together, PAN and PPN are generally

expected to account for 75-90% of observed $\sum PNs$ (Wooldridge et al., 2010) but here only account for 50% (Nault et al., 2024, Fig. 3e). Table 4 lists the PN species in the observations and the base model along with their main precursor and fraction relative to PAN. Figure 6a+b shows the speciation of $\sum PNs$ in the base model and with increased VOCs. The base model included two other higher PNs: MPAN formed from methacrolein and BZPAN formed from benzaldehyde. Each were 2% or less of PAN (Table 4). Figure 6a illustrates that the base model therefore had no ability to represent the larger fraction of higher

PNs compared to $\sum PNs$ in the observations. Alkyl nitrates (ANs), another product of VOC oxidation in the presence of $NO_x$,



also showed a large missing speciated fraction where individual measurements were only able to account for approximately one quarter of total observed $\sum ANs$ (Fig. S4a). More ANs were modeled than were measured but model $\sum ANs$ were still underestimated by 50% below 2km (Fig. 3f). Speciated model ANs for the model (with scaled VOCs and added PN chemistry discussed below) are given in Fig. S4b. The finding here of missing model $\sum ANs$ is similar to the 50% underestimate in Fisher

et al. (2016) in the U.S. and the 70% underestimate in Kenagy et al. (2021) during KORUS-AQ attributed to missing precursors and/or chemistry from non-biogenic precursors such as S/IVOCs. Here we simulate more ANs than in Kenagy et al (2021) partially due to our addition of the ALK6 alkylnitrate (R6N2, Table S1) described in Section 3 that had an average concentration of ~60 ppt below 2km (Fig. S4b).

(Lee et al., 2022) performed box modeling of VOCs during KORUS-AQ near the Daesan petrochemical complex (DPCC) to the southwest of Seoul and identified two higher PNs, peroxyhydroxyacetic nitric anhydride (PHAN) and peroxybenzoic nitric anhydride (PBZN), that contributed 17% and 6% to $\sum PNs$, respectively. They also identified an additional PN, peroxyacrylic nitric anhydride (APAN) that contributed up to 14% of $\sum PNs$ near the emission source. APAN, formed from 1,3-butadiene ($C_4H_6$) and acrolein, has been previously observed over petrochemical industrial areas (Roberts et al., 2001) and possibly in

more remote locations (Tanimoto and Akimoto, 2001). A PN formed from α-pinene oxidation through pinonaldehyde (PINPAN) was identified by (Noziere and Barnes, 1998) and estimated to be similar in concentration to MPAN. Nault et al. (2024) performed box-modeling using the MCMv3.3.1 in the F0AM box model to calculate potential additional PNs from the suite of observed VOCs during KORUS-AQ similar to Lee et al. (2022) but for the SMA. Identified PNs included PHAN, APAN, and PINPAN, as well as PNs from methyl ethyl ketone (MEKPN), limonene (LIMPAN), and aromatics (AROMPN).

We devised a chemical mechanism to produce these PNs for GEOS-Chem which is provided in Table S1 and Table S2.

Figure 6c shows the model results with scaled VOCs and a revised scheme for producing additional PNs (listed in Table 4). APAN is less important in the SMA (<1% of PAN) than in the results of Lee et al. (2022) which used observations closer to the source of $C_4H_6$ emissions (Daesan chemical complex). We found that 4 out of the 6 added higher PNs had a ratio to PAN

at least as large as MPAN (2%), with PHAN having the largest ratio (9%) followed by PINPAN (6%). PAN itself decreased by 8% in the revised scheme, which we attribute to both the re-speciation of the PAN acetylperoxy radical (CH3CO3) into other acetylperoxy radicals such as the 2-hydroxyacetylperoxy radical (GCO3) that makes PHAN (Table S2), and the removal of more peroxy radicals overall by the added higher PNs that would otherwise participate in ongoing photooxidation.

Figure 1c shows that the inclusion of additional higher PNs reduced ozone production by -2% against aircraft observations, with a net impact on ozone of -1 ppb which also occurs at the surface (Fig. 4a). This result agrees with the finding from Nault et al. (2024) that PNs were a net sink for ozone production in the SMA during KORUS-AQ. The addition of higher PNs reduced formaldehyde by -2% and acetaldehyde by -4% (Fig. 3a-b), which we attribute to the increased radical sink as modeled OH decreased by -5%. Figure 5c shows the impact on MDA8 ozone from adding modeled higher PNs to the simulation with



scaled VOCs (Fig. 5b). Ozone decreased across South Korea with the largest differences of -1 to -2 ppb in the SMA and Busan. Figure 7 shows PAN (a) and higher PNs (b) compared to formaldehyde. The PAN-formaldehyde relationship improved in the model with scaled VOCs and added PN chemistry, with the remaining bias in formaldehyde evident above approximately 6 ppb. The model underestimates the production of higher PNs as a function of formaldehyde which suggests that an additional VOC source of these species is needed.


We performed global simulations at 2×2.5° with the base model and the addition of higher PNs (without scaling VOCs in South Korea) to test the global relevance of these species. Figure 8 shows the global average surface concentrations from May 1 to June 10, 2016, of base model PNs (PAN, MPAN, PPN, BZPAN), newly added higher PNs (PHAN, LIMPAN, PINPAN, AROMPN, MEKPN, APAN), and the difference in the revised model $\sum PNs$ compared to the base model. The individual
concentrations of the added PNs are shown in Figure S5. Over land, $\sum PNs$ increased from +2 to +46% with the newly added higher PNs (Fig. 8c), with the largest increase over the Amazon from monoterpene-derived PNs (LIMPAN and PINPAN, Fig. S5). C4H6 emissions were only included in the KORUSv5 inventory and therefore APAN was only simulated in East Asia with a maximum in China. AROMPN only increased in the Northern Hemisphere due to higher aromatic emissions. Globally, in order of importance, the maximum concentration of PAN was 1.3 ppb followed by PPN at 0.5 ppb. The maximum
concentrations of PINPAN, PHAN, and LIMPAN were between 100 and 160 ppt. AROMPN, MPAN, BZPAN, and MEKPAN were between 20 and 70 ppt. APAN was negligible (2 ppt). All higher PNs are expected to be more important near source regions and APAN likely has greater relevance to local photochemistry, as in higher resolution studies such as in Lee et al. (2022). Over land, ozone was reduced by -1 ppb by the added PNs (Fig. 8d), with differences in $NO_2$ and OH of up to -20% and -5%, respectively. Smaller increases in ozone, $NO_2$, and OH were simulated over the ocean due to transport of $\sum PNs$.


Figure 9 shows the global average model fraction of more commonly measured PNs (PAN, MPAN, PPN) compared to $\sum PNs$ (PAN, MPAN, PPN, BZPAN, PHAN, LIMPAN, PINPAN, AROMPN, MEKPN, APAN) in the model with the new PN chemical scheme (Table S2). The global fraction is reduced from largely 100% to between 50% and 100% (up to a 50% reduction) with the largest change in regions dominated by monoterpene PNs (LIMPAN and PINPAN, Fig. S5). Over the
United States, Europe, and Asia, this fraction is reduced from approximately 100% to between 80 and 90%. Globally, PAN itself decreases similarly to the results shown for the SMA in Fig. 3d.

## 6 Evidence for unmeasured VOCs and their effects on ozone chemistry

Figure 3a shows that despite better simulating the suite of observed VOCs (Fig. S1), the model with scaled VOCs still underestimated formaldehyde in the SMA by -1.4 ppb (-20 %) below 2 km. Ozone, higher PNs, and $\sum ANs$ also remained
underestimated by -16 ppb (-17 %), -720 ppt (-57%), and -370 ppt (-52%) below 2 km, respectively. This implies that additional unmeasured VOCs must be present. Nault et al. (2024) used the observed relationship between $O_x$ and $\sum ANs$ in the SMA





during KORUS-AQ to estimate that there must be an average of 1.7 s$^{-1}$ of additional OH reactivity from unmeasured VOCs (shown on Fig. 1a). As in Nault et al. (2024) we calculated that this additional reactivity could further increase calculated ozone production by +2 ppb hr$^{-1}$ (Fig. 1c). This additional OH reactivity would likely bring the modeled OH overestimate at lower NO$_x$ (Fig. 2b) into better agreement with observations as observed by our sensitivity study with increased VOCs (Fig. 2d) and would further improve the model NO to NO$_2$ ratio (Fig. S3c).

Nault et al. (2024) suggested that likely sources of missing reactivity could be oxygenated VOCs, such as nonanal, and cycloalkenes/alkenes, such as monoterpenes. Nonanal is reactive with a lifetime against oxidation by OH of ~3 hrs for the conditions in the SMA (OH = 3E6 molecules cm$^{-3}$) and would produce both higher PNs and $\sum ANs$ (Bowman et al., 2003). Nonanal as well as octanal were recently measured at levels of 1 ppb or greater in several cities in the U.S. (Coggon et al., 2023b, a). In addition to other smaller carbonyls, such as formaldehyde and acetaldehyde, >C5 aldehydes such as nonanal are emitted from cooking activities (Ho et al., 2006; Coggon et al., 2023a). VOCs from cooking are minimally represented in emissions inventories but could have a mass contribution nearly as large as emissions from mobile sources (Coggon et al., 2023a). Similarly, anthropogenic monoterpene emissions are a poorly represented source in models that could make up as much as half of total monoterpenes even in a biogenically active area (Peng et al., 2022; Borbon et al., 2023; Peron et al., 2024). We hypothesize that OVOCs such as >C5 aldehydes or cycloalkenes/alkenes such as monoterpenes could be a potential source of unaccounted for OH reactivity in some cities and would contribute to resolving missing $\sum PNs$ (Fig. 3d) and $\sum ANs$ (Fig. 3f). These species could also help reconcile modeled formaldehyde (Fig. 3a) and ozone (Fig. 3c) and would improve the model OH overestimate at lower NO$_x$ levels (Fig. 2b).

## 7 Conclusions

Simulations of ozone pollution in urban areas, particularly those that are VOC-limited, rely on a successful representation of VOC emissions in the model inventory. These VOCs also produce organic NO$_x$ and radical reservoirs which serve as a local ozone sink but can produce ozone downwind. However, inventories of VOC emissions are more difficult to produce than for NO$_x$ given the larger number of compounds involved. Globally, VOC inventories have been shown to poorly represent local measurements (von Schneidemesser et al., 2023; Rowlinson et al., 2023). During the joint National Institute of Environmental Research (NIER) and National Aeronautics and Space Administration (NASA) Korea-United States Air Quality (KORUS-AQ) field study during May and June 2016, models underestimated ozone, formaldehyde, $\sum PNs$, and $\sum ANs$ in the Seoul Metropolitan Area. This points to underestimated VOCs in the emissions inventory as the regional photochemistry is VOC-limited.

We assessed average model biases in observed VOCs and we increased emissions estimates in South Korea to improve model agreement. Large model scale factors were required to reproduce observations for species related to volatile chemical products




(methanol, acetone, monoterpenes, methyl ethyl ketone, xylenes, ethylbenzene), LPG and natural gas emissions (propane,
butanes), and long-range transport (CO, $C_2H_2$, benzene). We expect that scale factors were overestimated given that we only
considered underestimated VOCs in South Korea and not upwind in East Asia where we had minimal constraints on emissions.
Scaling modeled VOC emissions individually to better match observations resulted in an increase in modeled OH reactivity
from 4.9 to 7.3 $s^{-1}$ and an increase in modeled average $PO_x$ by +2 ppb $hr^{-1}$. Scaled VOC emissions improved formaldehyde by
+30% and acetaldehyde by +120%. Ozone increased by +6 ppb aloft and up to +9 ppb at the surface. We found that ethanol
emissions were largely responsible for the improved model acetaldehyde and had a similar impact as isoprene or alkenes on
ozone production. Therefore, ethanol emissions may be important to consider in policy decisions regarding VOC reductions.

Peroxyacetyl nitrate (PAN) is produced in the SMA from many of the VOCs that were scaled to match observations including
ethanol. This scaling improved the modeled PAN underestimate from -50% to -23%. We added model chemistry to produce
six additional acyl peroxy nitrates (PNs) and found that four were at least as abundant as MPAN (2% of PAN. Efforts should
be made to look for these species (PHAN, LIMPAN, PINPAN, AROMPN) in future measurement campaigns. The addition
of these species reduced the fraction of commonly measured PNs (PAN, PPN, MPAN) globally from 100% in the base model
to between 50 and 100% depending on location. Over continents, the additional PNs reduced ozone by -1 ppb and OH by -
5%.


Remaining model underestimates below 2 km in formaldehyde (-1.4 ppb, -20%), ozone (-16 ppb, -17%), higher PNs (-720
ppt, -57%), and $\sum ANs$ (-370 ppt, -52%) are consistent with recent work finding that there is an average missing OH reactivity
of 1.7 $s^{-1}$ in the SMA (Nault et al., 2024). Likely sources of this reactivity, which would also produce formaldehyde, ozone,
PNs, and ANs, are OVOCs such as >C5 aldehydes (e.g., nonanal) from cooking emissions and anthropogenic sources of
cycloalkenes/monoterpenes such as limonene. Both VCPs and cooking emissions are poorly represented or missing from most
VOC inventories. Recent work focusing on these emissions and their chemistry (Coggon et al., 2023a; Warneke et al., 2023;
Peng et al., 2022) should greatly improve models' ability to simulate urban air quality.

**Code and Data Availability**

The GEOS-Chem model used here and the analysis codes are available at https://doi.org/10.5281/zenodo.10819248. The
F0AM box-modeling work from Nault et al. (2024) is available at https://doi.org/10.5281/zenodo.10723227. The KORUS-
AQ aircraft data is available at Chen (2018).



## Author contribution

BAN and KRT conceptualized the manuscript. BAN performed the formal analysis of pO$_3$ with input from KRT and JHC. KHB developed the new chemistry scheme for monoterpenes and new peroxynitrates in collaboration with KRT and BAN. DRB, RCC, AF, SRH, LGH, YRL, SM, KEM, IJS, and KU collected the observations used in this research and guided their use in this manuscript. KRT performed the modeling and analysis of results. All co-authors assisted with review and editing of the manuscript.

## Competing interests

At least one of the (co-) authors are members of the editorial board of Atmospheric Chemistry and Physics.

## Acknowledgements

BAN and KRT acknowledge NASA grant 80NSSC22K0283. SRH and KU were supported by NASA grant NNX15AT99G. DRB, SM and IJS acknowledge NASA grant NNX15AT92G. We acknowledge William H. Brune, Alexander B. Thames, and David O. Miller for their measurements of OHR, OH, and HO$_2$. We acknowledge Paul Wennberg and John Crounse for the measurements from CIT-CIMS. We acknowledge Glenn Diskin and Joshua DiGangi for their measurements of CO and CH4. We acknowledge Andrew Weinheimer for his measurements of O$_3$, NO, NO$_2$ and Armin Wisthaler for his measurements of VOCs. We thank Tori Barber for her helpful discussions.

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

**Table 1**. Description of the ground site and aircraft observations used in this work.[1]

| Instrument | PI | Measured species used in this work | Reference[2] |
|---|---|---|---|
| **Ground Observations** | | | |
| *Olympic Park*[3] | | | |
| 2B Tech 211, Teledyne T200U, Teledyne T500U CAPS, Aerodyne QCL | James Szykman and Andrew Whitehill | $O_3$, $NO_2$ | N/A |
| Dasibi Model 2108 Oxides of Nitrogen Analyzer | NIER | $O_3$, $NO_2$ | N/A |
| **DC8 Aircraft** | | | |
| Caltech CIMS (CIT-CIMS) | Paul Wennberg | glycolaldehyde, C5O3H10, C3O3H6, cresol, phenol, glycolaldehyde, hydroxyacetone, CH3OOH, peroxyacetic acid, hydroxynitrates | St. Clair et al., (2010; Crounse (2006) |
| Proton-transfer-reaction time-of-flight mass spectrometer (PTR-ToF-MS) | Armin Wisthaler | acetaldehyde, methanol, acetone, monoterpenes, benzene, toluene, methyl ethyl ketone | Tomsche et al. (2023) |
| Compact Atmospheric Multispecies Spectrometer (CAMS) | Alan Fried | formaldehyde | Richter et al. (2015) |
| Airborne Tropospheric Hydrogen Oxides Sensor (ATHOS) | William Brune | OH | Faloona et al. (2004); Brune et al. (2020) |



| NCAR 4-Channel chemiluminescence instrument | Andrew Weinheimer | O₃, NO, NO₂ | Weinheimer et al. (1993, 1994) |
|---|---|---|---|
| Georgia Tech–Chemical Ionization Mass Spectrometer (GT-CIMS) | L. Greg Huey | PAN, PPN | Slusher (2004); Lee et al. (2020) |
| Diode laser spectrometer (Differential Absorption Carbon monOxide Measurement, DACOM) | Glenn Diskin | CO, CH4 | Sachse et al. (1987) |
| Thermal Dissociation–Laser-Induced Fluorescence (TD-LIF) | Ronald Cohen | ΣANs, ΣPNs | Wooldridge et al. (2010); Day et al. (2002) |
| Whole Air Sampler (WAS) | Donald Blake | H₂, 1,3-butadiene, butenes, styrene, propene, isoprene, ethene, xylenes, ethyne, ≥C₂ alkanes, aromatics, halocarbons, alkyl nitrates | Simpson et al. (2020) |
| NCAR Charged-coupled device Actinic Flux Spectroradiometers (CAFS) | Samuel R. Hall | *j*-values | Shetter and Müller (1999) |

[1]For a full description of all KORUS-AQ observations, see Crawford et al. (2021).

[2]For specific measurement descriptions including uncertainty information, see the KORUS-AQ data archive (doi: 10.5067/Suborbital/KORUSAQ/DATA01)

[3]Olympic Park site in Seoul, 37.522°N,127.124°E

**Table 2.** Speciation of SAPRC99[1] for GEOS-Chem.[2]

| SAPRC99 | Base Model | New chemistry |
|---|---|---|
| ALK1 | C2H6 | C2H6 |
| ALK2 | 59% C3H8, 41% C2H2 | 59% C3H8, 41% C2H2 |
| ALK3 | ALK4 | ALK4 |
| ALK4 | ALK4 | ALK4 |
| ALK5 | ALK4 | ALK6 |
| ARO1 | 10% BENZ, 90% TOLU | 7% BENZ, 83% TOLU, 10% EBZ |
| ARO2 | XYLE | 63% XYLE, 37% TMB |
| OLE1 | PRPE | PRPE |
| OLE2 | PRPE | 7% STYR, 5% C4H6, 88% PRPE |
| TRP1 | MTPA (apinene, bpinene, sabine, carene) | MTPA |

[1]Definitions of SAPRC99 species are given in (Carter, 1999). [2]Definitions of GEOS-Chem species are given in species_database.yml in (Community, 2022), with some details in Table 3 and Table S1.


**Table 3**. Scale factors for modeled anthropogenic VOC and CO emissions.

| Species | Full name | Scale Factor |
|---|---|---|
| ACET | Acetone | 85x |
| ALK4 | ≥C4 alkanes | 3x |
| BENZ | Benzene | 2.4x |
| C2H2 | Acetylene | 2.5x |
| C3H8 | Propane | 9x |
| CO | Carbon monoxide | 3.6x |



| EBZ | Ethylbenzene | 2.1x |
|------|-------------|------|
| EOH | Ethanol | 40x |
| MEK | Methyl ethyl ketone | 70x |
| MOH | Methanol | 650x |
| MTPA | Monoterpenes | 450x |
| TMB | Trimethylbenzene | 0.32x |
| TOLU | Toluene | 1.3x |
| STYR | Styrene | 5x |
| XYLE | Xylenes | 1.5x |

**Table 4.** Model PN species descriptions, precursor, and observed and modeled PAN fraction.

| Model PN species | Full name | Main precursor | % of PAN Modeled (Observed) |
|------|-----------|---------------|------|
| **In standard model** | | | |
| PPN | Peroxypropionyl nitrate | > C3 aldehydes (RCHO) | 24% (6%) |
| MPAN | Peroxymethacroyl nitrate | Methacrolein (MACR) | 2% |
| PBZN | Peroxybenzoyl nitrate | Benzaldehyde (BALD) | 2% |
| **Added to model** | | | |
| APAN | Peroxyacrylic nitric anhydride | Acrolein (ACR) | 0.1% (1%) |
| AROMPN | Lumped aromatic PN | Lumped furanones (TLFUONE) | 4% |
| PINPAN | $\alpha$-Pinonyl peroxynitrate | Pinonaldehyde (PINAL) | 6% |
| LIMPAN | Limononyl peroxy nitrate | Limonaldehyde (LIMAL) | 2% |
| MEKPN | Hydroxypropanonyl peroxy nitrate | Methyl ethyl ketone (MEK) | 1% |
| PHAN | Peroxyhydroxyacetic nitric anhydride | Glycolaldehyde (GLYC) | 9% |


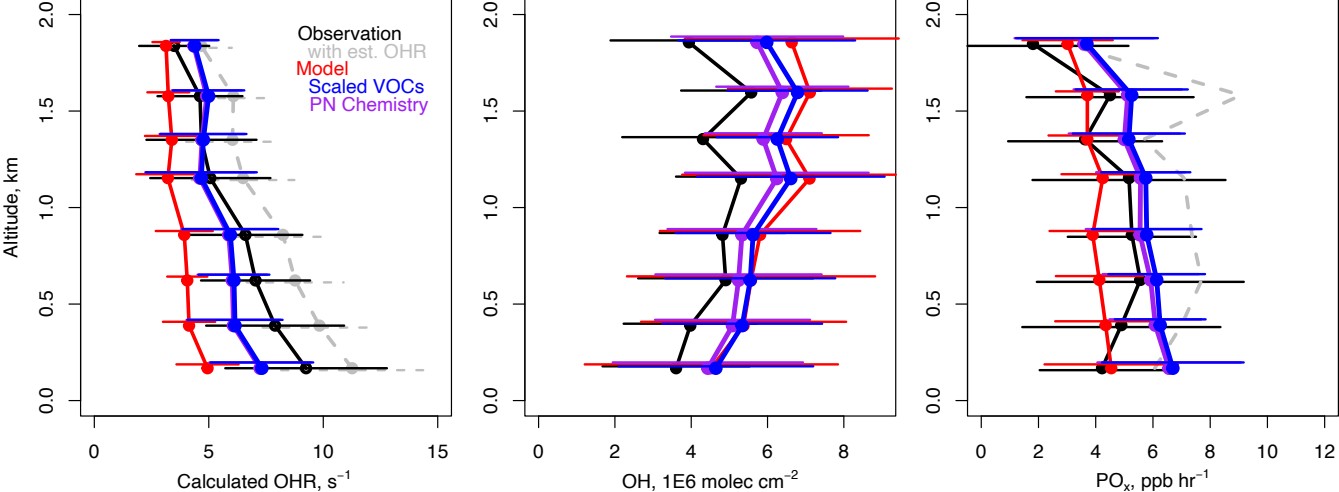

**Figure 1** – Mean vertical profile and standard deviation in the Seoul Metropolitan Area (SMA) (127.1 to 127.7ºE, 37.2 to 37.7 ºN) from May 1 to June 10, 2016, for data collected after 11am local time for a) calculated OH reactivity (OHR) for VOCs +





CO (Table 1), b) OH, and c) net production of $O_x$ ($PO_x=O_3 + NO_2$). Data are averaged into 8 bins of approximately 240 m for

the observations (black), the base model (red), the model with scaled VOCs (blue), and improved peroxynitrate (PN) chemistry

(purple). Calculation of OHR, the inclusion of estimated missing OHR (est. OHR), and net $PO_x$, and descriptions of model

simulations are given in Sections 3 and 4.

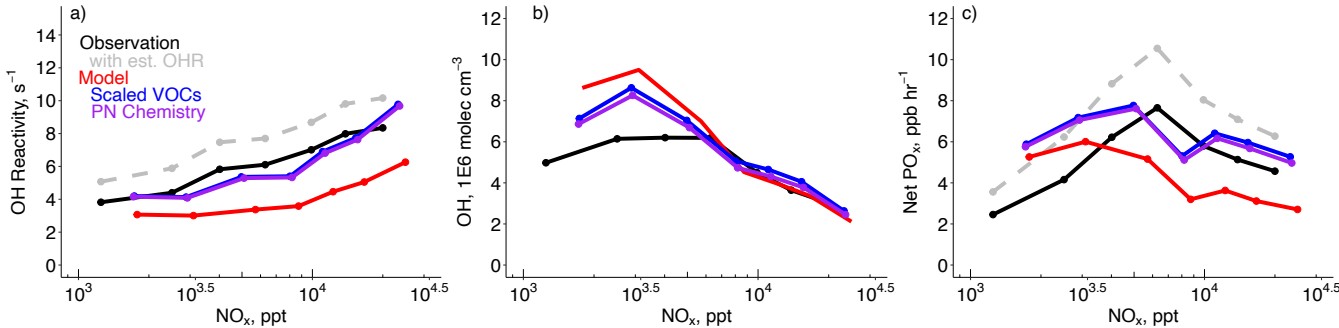


**Figure 2** – Same as Fig. 1 but for a) calculated OH reactivity for VOCs + CO (Table 1), b) OH, and c) net production of $O_x$

($PO_x=O_3 + NO_2$) plotted against $NO_x$ concentrations.

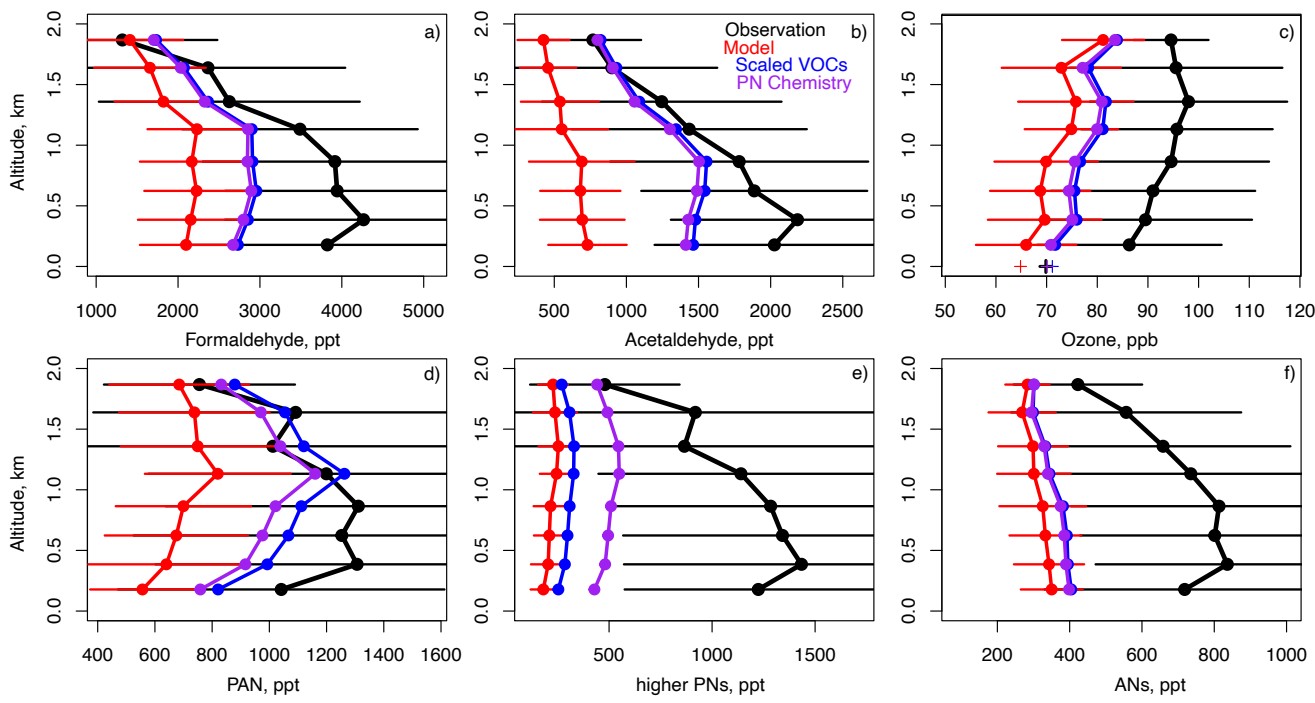

**Figure 3** – Same domain and model simulations as Fig. 1 but for a) formaldehyde, b) acetaldehyde, c) ozone, d) PAN, e)

higher PNs, and f) ANs. Surface ozone at Olympic Park between 11 and 16 local time during flight days is also plotted on

panel c) (+ symbols).



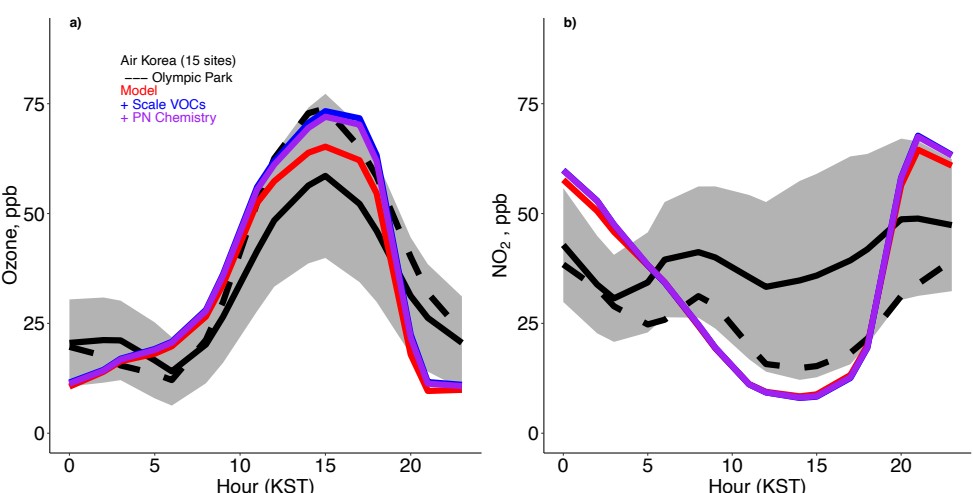

**Figure 4** – Mean diurnal cycle for a) ozone and b) NO$_2$ for the AirKorea sites within the GEOS-Chem grid box at Olympic
Park. The dashed line represents the EPA monitor at Olympic Park (Table 1). The gray shading represents the standard
deviation across the AirKorea sites (see Fig. 1b, Travis et al., 2022). KST is Korean standard time (UTC+9).

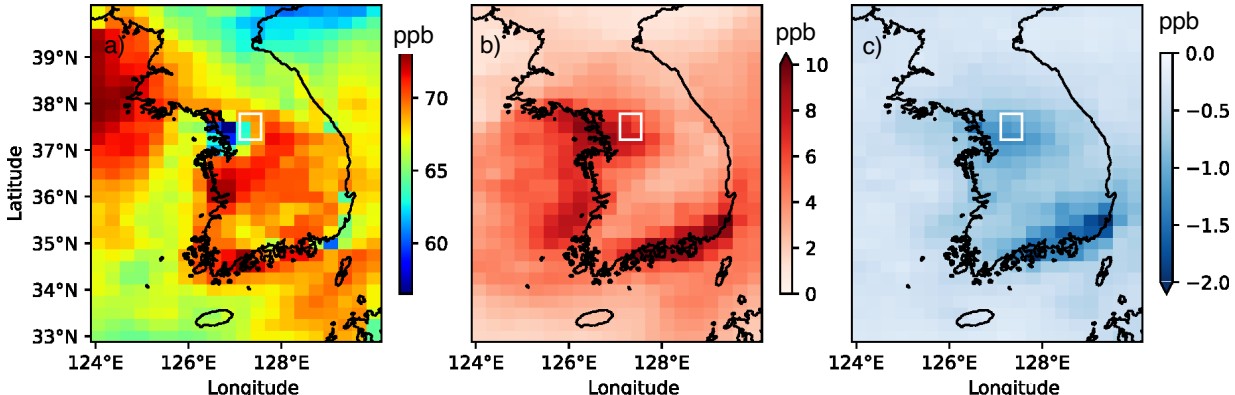

**Figure 5** – a) Maximum daily eight-hour average (MDA8) ozone from May 1 to June 10, 2016, and the impact from b)
scaled VOCs as described in Section 2 and c) adding peroxynitrate (PN) chemistry to the simulation in b) described in
Section 5. The white box designates the Seoul Metropolitan Area (SMA).



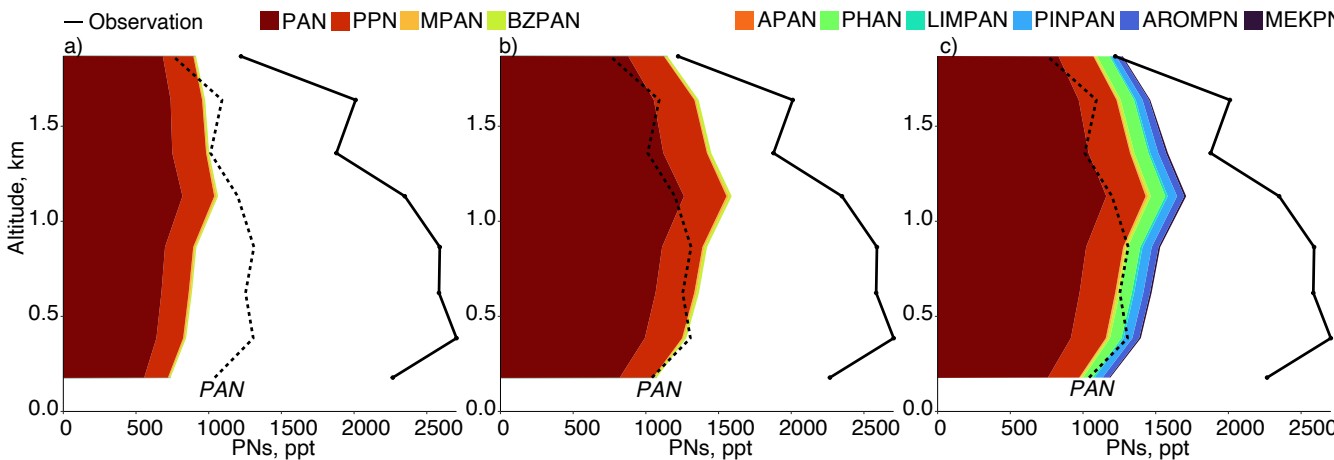

**Figure 6** – Speciated mean vertical profiles of modeled PNs for the domain of Fig. 1 compared against observed PNs (solid black line) and PAN (dashed black line) for a) base model, b) scaled VOCs, and c) added PN chemistry.

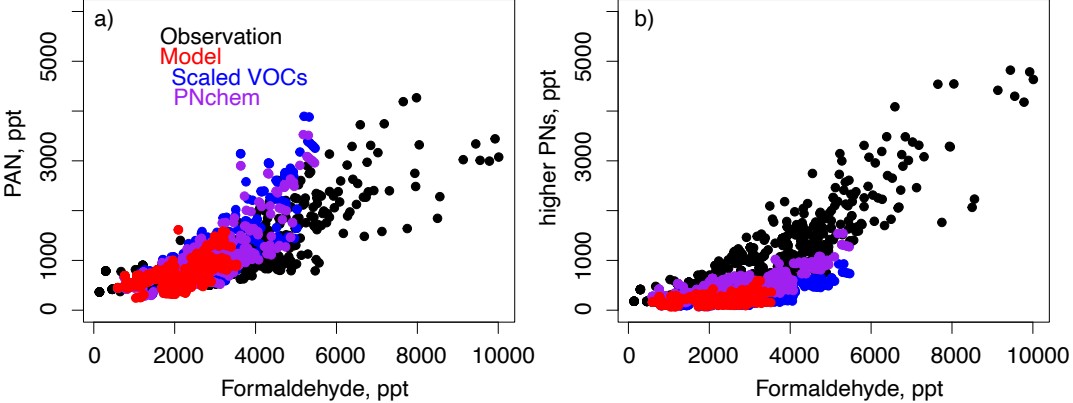

**Figure 7** – Comparison of a) PAN and b) higher PNs against formaldehyde for individual modeled and observed datapoints in the domain of Fig. 1. Model sensitivity studies for scaled VOCs and added PN chemistry are described in Section 2 and 5, respectively.





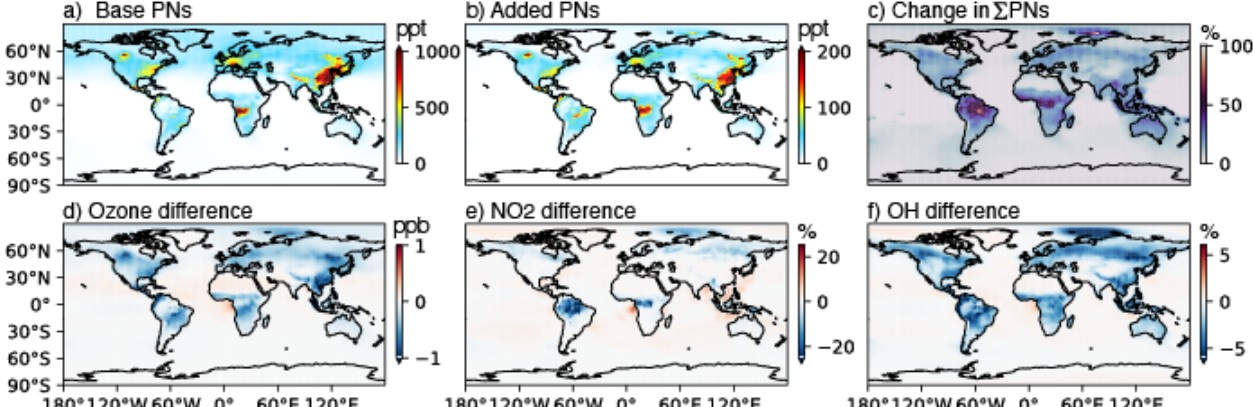

**Figure 8** – Global average surface concentrations at 2×2.5º from May 1 to June 10, 2016 for a) base PNs = PAN, MPAN, PPN, and BZPAN (Table 4), b) added PNs = PHAN, LIMPAN, PINPAN, AROMPN, MEKPN, APAN (Table 4), c) percentage change in the revised model (Base + added PNs) compared to the base model, d) difference in surface ozone, e) difference in surface NO2, and f) difference in surface OH.

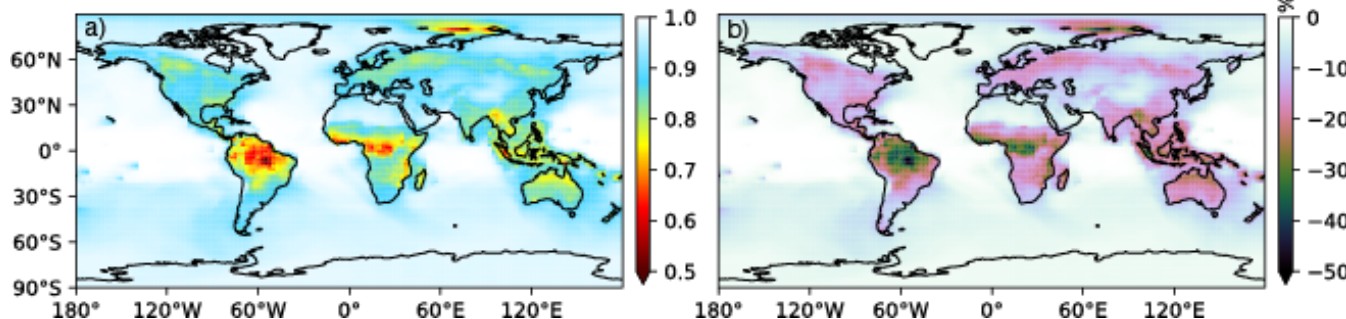

**Figure 9** – a) Modeled global average fraction of commonly measured PNs (PAN, MPAN, PPN) compared to $\sum PNs$ (PAN, MPAN, PPN, BZPAN, PHAN, LIMPAN, PINPAN, AROMPN, MEKPN, APAN). b) Reduction in commonly measured PNs vs. $\sum PNs$ in the revised model simulation (Section 5) compared to the base simulation (approximately 100%).