# Peer review of "Impact of improved representation of VOC emissions and production of NOx reservoirs on modeled urban ozone production"

_EGUsphere, 2024_

## Author Comment (AC1)

**Response to comments on "Impact of improved representation of VOC emissions and production of NOx reservoirs on modeled urban ozone production."**

We thank the reviewers for comments and suggestions that have helped to improve and clarify our paper. Comments from reviewers are in black, responses are blue, and new text is bold blue.

**RC1**

This study aims to address the underestimation of volatile organic compounds (VOCs) in emission inventories by adjusting scaling factors, optimizing photochemical reactions, and revising the schemes for generating additional polycyclic aromatic hydrocarbons (PAHs). They utilized both ground-based and airborne observation data to evaluate the extent of the impact of improved VOCs emissions on the Seoul Metropolitan Area. This contributes to the enhancement of the model's ability to simulate urban air quality. This paper is exceptionally well-crafted, presenting intriguing findings. Consequently, I suggest it should be published following appropriate revisions.

We thank Reviewer 1 for their support for this paper.

1. To improve the alignment between observed VOC speciation and model predictions, the chemical mechanism has been revised and underestimated VOCs species has been increased. These revisions are designed to improve the understanding of VOC species in the atmosphere and to enhance the simulation capability of the model. I would like to know if these improvements are suitable for other regions than the SMA and the period other than May 1 to June 10, 2016.

We cite a significant amount of work on observed VCPs and cooking emissions from the United States as well as Asia. We added the following to clarify this on Page 14, line 439:

**"in urban regions around the world."**

We also added the following discussion on Page 14, line 440 to discuss future needs for this mechanism:
**Future work will evaluate the ability of the revised model chemistry here to simulate observations from the Atmospheric Emissions and Reactions Observed from Megacities to Marine Areas (AEROMMA) field study in the United States during June to August 2023 (https://csl.noaa.gov/projects/aeromma/), and the Airborne and Satellite Investigation of Asian Air Quality (ASIA-AQ) field study (https://www-air.larc.nasa.gov/missions/asia-aq/index.html) from January to March 2024."**

Finally, we believe that this chemistry scheme will improve modeling of fire-related peroxynitrates. We changed the reference from remote regions in the introduction to the following reference to fires to help introduce this idea on Page 10, line 326.

**"and in fire plumes (M. Roberts et al., 2022)."**

We also added the following discussion on recent studies that found APAN in fire plumes on page 13, line 428.

**Species found to be less important during KORUS-AQ here, such as APAN, have been detected in larger amounts in studies of fire plumes (Decker et al., 2019; M. Roberts et al., 2022). While we do not consider fire emission precursors to APAN such as acrolein in this work here, the chemistry scheme provided for this PN species will aid in modeling studies of the impacts of fire emissions on atmospheric composition.**

We also added a reference to fires in the abstract on page 2, line 43

**"campaigns as well as considered from other VOC emission sources (e.g., fires)"**

2. Line 158, Are you referring to the process of individually adjusting the scaling factors for VOCs species and comparing them with observed values to determine the optimal scaling factors? And the determination of the best

scaling factor typically involves assessing how well the adjusted model outputs align with actual measurements. What metrics were used in this study to quantify the differences between predicted and observed values?

RC2 also asked about the derivation of the scale factors. To address both comments:

Yes, we are referring to individually adjusting the scaling factors for VOCs species and comparing them with observed values to determine the optimal scaling factors.

We added a column to the Supplement Table S3 that now gives the average base model bias and scaled VOCs bias below 2km. We also added the following text to page 5, line 67:

"We applied scaling factors to the individual KORUSv5 VOC emissions over South Korea until modeled concentrations **matched average observations below 2km within 30% or better (Table S3). The one exception is trimethylbenzenes where we improved the model bias from +93% to only -43%, but the absolute bias was < 10 ppt."**

3. Line 219, The observations clearly show a shift from increasing to decreasing POx with increasing NOx at approximately 6 ppb. However, the model did not capture this feature at all. What could be the reason for this? In addition, many discrepancies between simulations and observations in this study have been explained by insufficient model resolution. Could this be further addressed through the use of high-resolution models?

Yes, we meant to state that this could be due to insufficient resolution. We clarified this with additional text on Page 7, line 231. "We hypothesize that model difficulty in capturing **the behavior of PO$_x$ and the separation** of these two regimes is also due to insufficient model resolution."

We agree that many discrepancies can be explained by insufficient resolution and we add the following sentence to summarize the benefits of future work with increased resolution at the end of the conclusions on Page 14, line 443.

**"We find that to best simulate these studies, models with higher resolution that used here (~25km) would better be able to capture the behavior of OH and PO$_x$ at varying levels of NO$_x$, but resolution alone will not resolve model underestimates in HCHO and OH reactivity without improvements to VOC emissions inventories."**

4. After scaling VOCs, the model simulations of NOx concentrations tend to be underestimated during the day and overestimated at night in Fig. 4b. Could you provide a more detailed explanation for this phenomenon?

This is true before and after scaling as shown on Fig. 4b. We added in the following sentence on page 9, line 280, to address the overestimate at night:

**"Nighttime model overestimates in NO₂ are likely attributable to an overly shallow nighttime mixed layer height which results in excessive ozone titration (Travis et al., 2022)."**

5. The model does not fully capture the vertical profiles of PNs in the observation in Fig. 6. Is this due to an underestimation in the VOC emissions?

Yes, we state this on Page 12, line 378. "Ozone, higher PNs, and $\sum ANs$ also remained underestimated by -16 ppb (-17 %), -720 ppt (-57%), and -370 ppt (-52%) below 2 km, respectively. This implies that additional unmeasured VOCs must be present."

6. Line 370, Please verify the absence of Figure 2d in the article.

Thank you for catching this. I believe this was a second extraneous reference to Fig. 2b and thus I have removed it.

7. Line 230 The word "tha" appears to be a spelling error.

Fixed.

8. Figure 1 lacks a serial number indicating its order.

Fixed.

**RC2**
Travis et al. examine the impact of modified representation of VOC emissions and production of NOx reservoirs on modeled urban ozone production. The modifications are largely informed by the interpretation of KORUS-AQ observation campaign. Their findings reveal that significant augmentation of anthropogenic emissions is necessary to align with the observed VOC species and calculated OH reactivity. The enhanced VOCs also partly reduced the underestimation of PAN, but the underestimation of PNs is not resolved. The results emphasize the need for improved measurement and research into OVOCs emissions and chemistry to enhance the model's capacity to replicate PNs and ANs.

The study is surely significant and aligns with the scope of ACP. Although I do not have specific criticisms regarding the data analysis and interpretation, I do have two primary concerns that I would like the authors to address before I can recommend the manuscript for publication.

We thank the reviewer for the thoughtful reading of the paper.

First, the modification to emission inventory and "improved" chemistry scheme is largely relies on the KORUS-AQ campaign. In the abstract, the authors state that the study is motivated the fact that "during KORUS-AQ, air quality models underestimated ozone, formaldehyde, and peroxyacetyl nitrate (PAN) indicating an underestimate of VOCs in the emissions inventory". However, the GEOS-Chem model has been extensively applied to diverse regions globally, each with distinct chemical environments. For example, numerous studies have demonstrated that the standard version of the GEOS-Chem model (particularly versions after 12.0.0) tends to overestimate surface ozone concentrations in other polluted regions, such as China. This implies that the new chemistry may not fit other regions. My primary concern is whether the new chemical schemes introduced in this study, which might be integrated into the standard GEOS-Chem scheme but heavily rely on the findings from the KORUS-AQ campaign, will have universality for other regions. This issue needs to be evaluated before the scheme can be considered for adoption in GEOS-Chem. Otherwise, these improvements may appear to be tailored solely for the region constrained by KORUS-AQ studies.

Without the reviewer providing citations for this statement "numerous studies have demonstrated that the standard version of the GEOS-Chem model (particularly versions after 12.0.0) tends to overestimate surface ozone concentrations in other polluted regions, such as China", we will respectfully disagree based on our own knowledge. We edited the citation below (Page 3, lines 72-73) to Gaubert et al., 2020 (https://doi.org/10.5194/acp-20-14617-2020) to clarify that this study shows showing large model underestimates in both Korea from KORUS-AQ and also China during the ARIAS campaign.

Gaubert et al. (2020) found that persistent underestimates in modeled carbon monoxide (CO) in **both South Korea and China** were partially responsible for the modeled ozone underestimate **in both locations**.

We cite a significant amount of work on observed VCPs and cooking emissions from the United States as well as Asia. We added the following to clarify this on Page 14, line 439:

**"in urban regions around the world."**

We also added a reference to work published after our submission describing the need for more VOCs and CO in a model of both China and South Korea on page 3, lines 80-83.

**"Kim et al. (2024) also discussed the importance of detailed VOC observations to constraining ozone precursors, as errors in biogenic emissions in urban areas can result in better agreement with ozone observations for the wrong reasons. This study also found that modeled CO was underestimated over both China and South Korea and large increases were needed in both CO and VOCs to improve model simulations."**

We also added the following discussion on Page 14, line 440 to discuss future needs for this mechanism:
**Future work will evaluate the ability of the revised model chemistry here to simulate observations from the Atmospheric Emissions and Reactions Observed from Megacities to Marine Areas (AEROMMA) field study in the United States during June to August 2023 (https://csl.noaa.gov/projects/aeromma/), and the Airborne and Satellite Investigation of Asian Air Quality (ASIA-AQ) field study (https://www-air.larc.nasa.gov/missions/asia-aq/index.html) from January to March 2024."**

Finally, we believe that this chemistry scheme will improve modeling of fire-related peroxynitrates. We changed the reference from remote regions in the introduction to the following reference to fires to help introduce this idea on Page 10, line 326.

**"and in fire plumes (M. Roberts et al., 2022)."**

We also added the following discussion on recent studies that found APAN in fire plumes on page 13, line 428.

**Species found to be less important during KORUS-AQ here, such as APAN, have been detected in larger amounts in studies of fire plumes (Decker et al., 2019; M. Roberts et al., 2022). While we do not consider fire emission precursors to APAN such as acrolein in this work here, the chemistry scheme provided for this PN species will aid in modeling studies of the impacts of fire emissions on atmospheric composition.**

We also added a reference to fires in the abstract on page 2, line 43

**"campaigns as well as considered from other VOC emission sources (e.g., fires)"**

Figure 8 and related discussions are a nice way to present the global impact of the added PNs chemistry on global chemistry. Can you also show the overall impact on surface ozone due to the modification of chemistry?

We show the impact on surface ozone in Figure 8d.

Secondly, the scaling of anthropogenic VOCs emissions appears to be somewhat arbitrary (Lines 158-166). While I acknowledge that the most of the model bias towards VOCs should be attributed to the emission inventory, the magnitude of the scaling factor will impact the interpretation of the effectiveness of the modified chemistry. I look forward to a more rigorous consideration of the emission scaling factor.

RC1 also asked about the derivation of the scale factors. To address both comments:

We added a column to the Supplement Table S3 that now gives the average base model bias and scaled VOCs bias below 2km. We also added the following text to page 5, line 60:

"We applied scaling factors to the individual KORUSv5 VOC emissions over South Korea until modeled concentrations **matched average observations below 2km within 30% or better (Table S3). The one exception is trimethylbenzenes where we improved the model bias from +93% to only -43%, but the absolute bias was < 10 ppt."**

Minor comments.

Line 29: It is also important to note the version of GEOS-Chem model here.

Added.

Figure 5: suggest add the description to the title of the figure.

We respectfully disagree with this suggestion as it would be inconsistent with the rest of the figures and we prefer to have a detailed caption.

**Other changes**

We found small error in our plotting routine that changed several numbers by very small amounts (< 5%). See edits on Page 7, lines 207-210 and line 227 and updated figures. We also found that we were inconsistent in reporting values for the lowest model level vs. the average below 2km, and edited the text accordingly in Section 6 (Page 12, lines 377-378.

*References*

Decker, Z. C. J., Zarzana, K. J., Coggon, M., Min, K.-E., Pollack, I., Ryerson, T. B., Peischl, J., Edwards, P., Dubé, W. P., Markovic, M. Z., Roberts, J. M., Veres, P. R., Graus, M., Warneke, C., de Gouw, J., Hatch, L. E., Barsanti, K. C., and Brown, S. S.: Nighttime Chemical Transformation in Biomass Burning Plumes: A Box Model Analysis Initialized with Aircraft Observations, Environ. Sci. Technol., 53, 2529–2538, https://doi.org/10.1021/acs.est.8b05359, 2019.

Kim, K.-M., Kim, S.-W., Seo, S., Blake, D. R., Cho, S., Crawford, J. H., Emmons, L. K., Fried, A., Herman, J. R., Hong, J., Jung, J., Pfister, G. G., Weinheimer, A. J., Woo, J.-H., and Zhang, Q.: Sensitivity of the WRF-Chem v4.4 simulations of ozone and formaldehyde and their precursors to multiple bottom-up emission inventories over East Asia during the KORUS-AQ 2016 field campaign, Geoscientific Model Development, 17, 1931–1955, https://doi.org/10.5194/gmd-17-1931-2024, 2024.

M. Roberts, J., Andrew Neuman, J., S. Brown, S., R. Veres, P., M. Coggon, M., E. Stockwell, C., Warneke, C., Peischl, J., and A. Robinson, M.: Furoyl peroxynitrate (fur-PAN), a product of VOC–NO x photochemistry from biomass burning emissions: photochemical synthesis, calibration, chemical characterization, and first atmospheric observations, Environmental Science: Atmospheres, 2, 1087–1100, https://doi.org/10.1039/D2EA00068G, 2022.

Travis, K. R., Crawford, J. H., Chen, G., Jordan, C. E., Nault, B. A., Kim, H., Jimenez, J. L., Campuzano-Jost, P., Dibb, J. E., Woo, J.-H., Kim, Y., Zhai, S., Wang, X., McDuffie, E. E., Luo, G., Yu, F., Kim, S., Simpson, I. J., Blake, D. R., Chang, L., and Kim, M. J.: Limitations in representation of physical processes prevent successful simulation of $PM_{2.5}$ during KORUS-AQ, Atmos. Chem. Phys., 22, 7933–7958, https://doi.org/10.5194/acp-22-7933-2022, 2022.